# Evaluating firefly extinction risk: Initial red list assessments for North America

**Candace E. Fallon** [1,2] *, **Anna C. Walker** [2,3], **Sara Lewis** [2,4], **Joseph Cicero** [5], **Lynn Faust** [2,6], **Christopher M. Heckscher** [2,7], **Cisteil X. Pérez-Hernández** [2,8], **Ben Pfeiffer** [2,9], **Sarina Jepsen** [1,2]

**1** The Xerces Society for Invertebrate Conservation, Portland, Oregon, United States of America, **2** IUCN SSC Firefly Specialist Group, Gland, Switzerland, **3** New Mexico BioPark Society, Albuquerque, New Mexico, United States of America, **4** Department of Biology, Tufts University, Medford, Massachusetts, United States of America, **5** School of Plant Sciences, University of Arizona, Tucson, Arizona, United States of America, **6** Emory River Land Co., Knoxville, Tennessee, United States of America, **7** Department of Agriculture and Natural Resources, Delaware State University, Dover, Delaware, United States of America, **8** Instituto de Investigaciones en Ecosistemas y Sustentabilidad, Universidad Nacional Autónoma de México, Michoacán, México, **9** Firefly Conservation and Research, New Braunfels, Texas, United States of America

* candace.fallon@xerces.org

**Data Availability Statement:** All relevant data are within the manuscript and its Supporting information files.

## Abstract

Fireflies are a family of charismatic beetles known for their bioluminescent signals. Recent anecdotal reports suggest that firefly populations in North America may be in decline. However, prior to this work, no studies have undertaken a systematic compilation of geographic distribution, habitat specificity, and threats facing North American fireflies. To better understand their extinction risks, we conducted baseline assessments according to the categories and criteria of the International Union for Conservation of Nature (IUCN) Red List for 132 species from the United States and Canada (approximately 79% of described species in the region). We found at least 18 species (14%) are threatened with extinction (e.g. categorized as Critically Endangered, Endangered, or Vulnerable) due to various pressures, including habitat loss, light pollution, and climate change (sea level rise and drought). In addition, more than half of the species (53%) could not be evaluated against the assessment criteria due to insufficient data, highlighting the need for further study. Future research and conservation efforts should prioritize monitoring and protecting populations of at-risk species, preserving and restoring habitat, gathering data on population trends, and filling critical information gaps for data deficient species suspected to be at risk.

## Introduction

Effective conservation planning and action depends on identifying the most at-risk species based on their estimated probability of extinction. The International Union for Conservation of Nature (IUCN) Red List of Threatened Species is considered the global standard for estimating the risk of species extinction and can be used as a first step in conservation efforts [1,2]. First established in 1964, major gains have been made in adding new assessments to the Red List in recent years, moving ever closer to the group's goal of 160,000 assessed species.

**Funding:** CEF and SJ were funded by the Samuel Freeman Charitable Trust, the Edward Gorey Charitable Trust (https://edwardgorey.org/), the New-Land Foundation (http://newlandfoundation. org/), Morningstar Foundation (http:// themorningstarfoundation.com/), and Xerces Society members. ACW was funded by the New Mexico BioPark Society (https://bioparksociety.org/ main/). The funders had no role in study design, data collection and analysis, decision to publish, or preparation of the manuscript.

**Competing interests:** The authors have declared that no competing interests exist.

Currently, the Red List comprehensively covers charismatic vertebrates, including mammals (91% of all species assessed) and birds (100% of species assessed) [3]. Invertebrates, in contrast, are profoundly underrepresented on the Red List, with just 2% of described species (24,219 out of an estimated 1,478,938) assessed as of 2020 [3]. This gap is even wider for insects: although they represent an estimated 53% of described animal and plant species, only 1% have been assessed [3].

Beetles, a hyper-diverse group of insects with an estimated 386,500 described extant species worldwide [4] have been identified as a priority group for Red Listing due to their species richness, assessment practicality (e.g., relatively stable taxonomy, adequate information available), and economic value [5]. The firefly beetles (family Lampyridae), which contain some 2,200 species globally [4], represent an ideal group for Red List assessments because these charismatic and cosmopolitan insects have the potential to serve as flagship species for invertebrate conservation. They possess diverse life history traits and behaviors and have been the subject of active evolutionary, behavioral, and genetic research [6–10]. Through biomedical research, firefly luciferase has facilitated numerous scientific advances [e.g., 11]. Furthermore, fireflies are culturally, ecologically, and economically important, and because of their sensitivity to light pollution and other environmental degradation, they may be important bioindicators of ecosystem health [12–17]. Some species have been used as biological control agents of unwanted land snails [18].

Long-term surveys have revealed local population declines of the glow-worm *Lampyris noctiluca* in the U.K. [19,20] and the congregating mangrove firefly *Pteroptyx tener* in Malaysia [21,22]. In North America, population declines have been anecdotally reported [16], but IUCN Red List assessments had yet to be conducted for any firefly species. A recent review of global threats to firefly persistence revealed habitat degradation and loss, light pollution, pesticide use, poor water quality, climate change, and invasive species to be among the major suspected drivers of decline [23]. Firefly tourism, which has increased rapidly in recent years and has been identified as a potential threat, offers an opportunity to examine how human activities can affect fireflies and their habitats, while determining how these activities can continue without causing local extirpations [17]. With emerging evidence for widespread declines in insect populations [24–26], there is an urgent need for formal assessments to inform the conservation status of firefly species and estimate their extinction risk.

This study summarizes global IUCN Red List assessments for fireflies in the U.S. and Canada, presenting the first formal estimates of extinction risk conducted for any member of this beetle family. We compiled available information on distributions, habitats, life history traits, behaviors, and threats for most (79%) of the currently described firefly species in the U.S. and Canada. Our goal in compiling this baseline data was to identify species at greatest risk of extinction, propose strategies for conserving threatened species, and highlight targets for future research.

## Methods

### Study organism

Fireflies (Coleoptera: Lampyridae) are holometabolous insects that spend the majority of their lives as larvae–sometimes up to 2 years or more–whereas adults may live only a few weeks [27]. Generation time and seasonality vary considerably depending on latitude, elevation, degree day range, and sex and species-specific emergence timing, in addition to weather and climate [28]. In general, generation time increases with higher latitudes and elevations. Southern fireflies may have one-year life cycles, whereas northern populations could have two to three-year life cycles [28]. However, because fireflies are facultative in their development time,

this period may increase in response to environmental variables such as drought [28] or increases in elevation (L. Buschman pers. comm. 2020). Similarly, the breeding season may be longer (year-round for some species) at southern latitudes, while it will be much shorter at higher latitudes or elevations (lasting only a week to a few months) [28].

As larvae, fireflies are voracious predators of soft-bodied invertebrates including snails, slugs, and worms [9], but may also be scavengers of dead insects and berries [29]. They are typically subterranean or found on or near the soil surface, in leaf litter, or in rotting logs, depending on the genera and/or species [16,28,30]. Adults of most species are not known to feed, although some species have been observed nectaring on flowers, mouthing leaves, and feeding on sap [28,31–34], and the females of some *Photuris* species are predatory mimics of other fireflies [35,36].

Although fireflies are known for bioluminescence, the actual bioluminescent capabilities of the group as a whole are these: the larvae of all known firefly species are luminescent [9], yet not all adults are capable of producing light. In the U.S. and Canada, fireflies can thus be organized into groups based on their bioluminescent capabilities: those that use flashing or glowing courtship signals (flashing fireflies and glow-worms), and those that do not (daytime dark species; in this context, 'dark' refers to non-luminescent or faintly luminescent diurnal species). Flashing fireflies, also known as lightningbugs, are typically crepuscular or nocturnally active; male and female adults use precisely timed flashes or flickers to communicate with potential mates [9]. Glow-worms are active during a similar time period but differ in that adult female glow-worms are typically flightless because their wings are short or even absent [9]. Furthermore, it is primarily the adult females that are luminescent, glowing to attract often non-luminescent males that fly overhead in search of a mate (there are some exceptions to this, e.g. *Phausis reticulata*) [9,37]. Daytime dark fireflies are diurnally active and are known [38] or suspected to use pheromones to locate potential mates [6,8].

Fireflies require moist conditions to prevent desiccation of larvae and their prey [9,16]. In general, fireflies are found in diverse habitats, including riparian woodlands, desert canyons, and coastal salt marshes. While some species are strict habitat specialists, others utilize a variety of habitats. Certain species opportunistically occupy urban and rural areas such as residential lawns, crop fields, and overgrown lots.

## Species checklist

We compiled a checklist of all native described species and subspecies of Lampyridae found in the U.S. and Canada based on Lloyd [39], which we updated to include recent species descriptions [30,40–42]. This yielded 167 species in 20 genera (S1 Table). Thirty-nine of these species were described in just the last 15 years [30,40–42], supporting speculation that as many as 225 species could occur in the U.S. and Canada [9]. One introduced European species, *Phosphaenus hemipterus*, reported from Nova Scotia [43], was not included. Synonymy was addressed using ITIS [44], Cicero [45], and other taxonomic references, where relevant. The updated checklist was reviewed by firefly experts (S2 Table). Thirty-five *Photuris* species that were described in 2018 [30] were excluded, as such recent description led to a paucity of data and lack of knowledgeable taxonomic experts, yielding a total of 132 taxa that were assessed.

## Literature review and data compilation

At the outset of the assessment process, we reviewed published literature and unpublished reports and solicited input from taxonomic experts. For 130 species and two subspecies, we compiled information on taxonomy, distribution, population size, ecology, behavior, threats, and any known conservation measures. We searched for relevant literature using Web of

Science, Google Scholar, the world Lampyridae literature database [45], and Fireflyers International Network's list of scientific and popular literature [46] using key words such as Lampyridae and individual species and genus names. Occurrence records were obtained from online biodiversity databases and museum collections (e.g., GBIF, SCAN, California Academy of Sciences), scientific literature, and species experts. Data were screened for anomalous records, which were vetted and removed if questionable. Unless pertaining to widespread or common species, observations from iNaturalist and BugGuide community science sites were only used if they had been verified by a taxonomic expert (Role = Determiner, see S2 Table). In some cases, records from the published literature were georeferenced in order to draft more detailed distribution maps.

## IUCN Red List methodology

We evaluated extinction risk for each species using the IUCN Red List Categories and Criteria: Version 3.1 [47]. Each species was assessed against five criteria with quantitative thresholds, which are based on standard biological indicators that render populations more vulnerable to extinction: A (past, present, or future population size reduction), B (geographical range size with evidence of decline, fragmentation, or fluctuation), C (small population size with decline, fragmentation, or fluctuation), D (very small or restricted population size), and E (quantitative analysis of extinction risk).

Depending on which criteria thresholds were met, each taxon was assigned to one of the following IUCN Red List categories: Extinct (EX), Extinct in the Wild (EW), Critically Endangered (CR), Endangered (EN), Vulnerable (VU), Near Threatened (NT), Least Concern (LC) or Data Deficient (DD). Species assigned to the categories CR, EN, or VU are considered threatened because they are facing extremely high, very high, or high risk of extinction in the wild, respectively. Species assessed as Near Threatened are close to qualifying for a threatened category and therefore may qualify as threatened in the near future. Species assessed as Least Concern are generally widespread and abundant and do not qualify for a threatened category under any of the criteria. A taxon is considered Data Deficient when there is not enough information on the distribution or population size to make a direct or indirect assessment of its extinction risk. Species are assessed as Extinct when there is no reasonable doubt that the last individual has died, and species are assessed as Extinct in the Wild when they are known to survive only in captivity [48]. Like many invertebrates, none of the species assessed had sufficient information on population size or rates of population size reduction to be evaluated against the thresholds for Criteria A, C, D and E. Therefore, all species assessed as threatened were done so under Criterion B, which is based on restricted ranges with evidence of decline, severe fragmentation, or extreme fluctuation in distribution or population size. Further details on the Red List methodology can be found in S1 File and the IUCN Red List Guidelines [48].

## Synthesis and review

Throughout the process, species experts (S2 Table) were consulted to verify that each species assessment and distribution map included accurate and up-to-date information. The majority of assessments (128 species) were published on the IUCN Red List in March 2021 [49], while the remaining four species are awaiting publication.

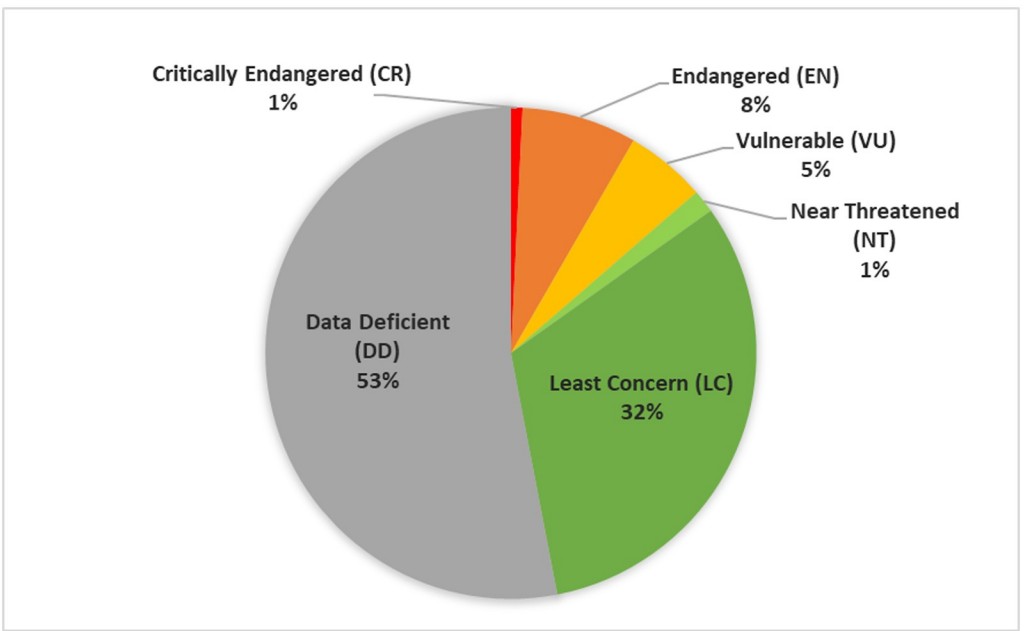

**Fig 1. IUCN Red List categories for 132 North American firefly species.**

## Results and discussion

### Extinction risk and threats

Our assessments suggest that at least 14% of evaluated North American firefly species (18 species) are threatened, classified as either Critically Endangered (CR, n = 1), Endangered (EN, n = 10), or Vulnerable (VU, n = 7) (Fig 1). In addition, 2% were categorized as Near Threatened (NT, n = 2) while 32% were classified as Least Concern (LC, n = 42). Over half (53%) of assessed firefly species were categorized as Data Deficient (DD, n = 70), which means there remains considerable uncertainty in the proportion of North American fireflies that may be at risk of extinction. Our estimate of 14% threatened likely represents a lower limit, with an upper limit of 67% should all DD species turn out to be threatened. Following methods used in other Red List assessments [50,51], if we assume that our Data Deficient species follow a pattern similar to those with sufficient data, we estimate that 29% (a "mid-estimate") of North American firefly species may eventually be classified as threatened (Table 1).

Invertebrate extinction risk has been linked to several different factors, including narrow geographical ranges, habitat specialization, and body size [52–54]. For fireflies, Reed et al. [55] identified risk factors expected to make species more susceptible to threats, including courtship activity period (nocturnal vs. diurnal), poor dispersal ability (due in part to adult female

**Table 1. IUCN Red List summary information for 132 North American firefly species.**

| Summary information | Count | Percentage |
|---|---|---|
| Total species evaluated | 132 | |
| Total species with sufficient data (CR+EN+VU+NT+LC) | 62 | 47% |
| Total Threatened—lower limit estimate (CR+EN+VU) | 18 | 14% |
| Total Threatened–mid estimate ((CR+EN+VU)/(total—DD)*total) | 38 | 29% |
| Total Threatened—upper limit estimate (CR+EN+VU+DD) | 88 | 67% |

brachyptery or aptery), and habitat specialization. In our assessments, these risk factors were found to be prevalent among firefly species with heightened extinction risk (Table 2).

For species with sufficient information to identify known and suspected threats to their persistence (88 species total), the primary threats included habitat loss and degradation, light pollution, and climate change and severe weather. Habitat loss has been identified as the biggest perceived threat to fireflies worldwide [23], rendering habitat specialists particularly vulnerable. All 18 species categorized as threatened are known or suspected to be restricted to specialized habitats like freshwater interdunal swales or cypress swamps (Fig 2), which makes them more vulnerable to habitat loss, degradation, and fragmentation. These threats are caused by a variety of human activities, including commercial and residential development, agricultural conversion, water pollution, groundwater pumping, waterway modifications, cattle grazing, and recreational activities such as off-road vehicle (ORV) use. Habitat loss and degradation can be particularly devastating for species with flightless females, which are more vulnerable to trampling or habitat destruction due to their limited dispersal capacity. Two of the species categorized as threatened, the Florida scrub dark firefly (*Lucidota luteicollis*) and the ant-loving scrub firefly (*Pleotomodes needhami*), have flightless adult females. Approximately a quarter of firefly species in the U.S. and Canada are known or expected to have flightless females, and 23 (68%) of such species were categorized as DD; thus, additional species are likely to eventually be categorized as threatened.

In general, moisture is critically important during all firefly life stages to prevent desiccation [9]; eggs, soft-bodied larvae, and flightless females may be particularly susceptible [60]. Thus, loss of moisture due to habitat manipulation, drought, or mismanagement of water resources can negatively impact fireflies. Because firefly larvae are predatory on soft-bodied invertebrates that are also susceptible to desiccation, loss of moisture can impact prey populations as well. Climate change is likely to be a major concern for many species. For example, in the arid American West, droughts are becoming more widespread, frequent, and severe due to a changing climate [61]. As a result of this, combined with changing precipitation patterns and increasing human demands, water tables are dropping [62,63], which can cause ephemeral aquatic habitats to go dry, interrupt flow regimes, and stress local plant communities [64]. Some western firefly habitats have completely disappeared due to water table reductions [65], and continuing declines in plant communities along riparian corridors in Texas are causing reduced moisture retention in the soil, which contributes to lower quality habitat for firefly larvae and diminishes the amount of water available to recharge aquifers (B. Pfeiffer pers. obs.).

Wetland habitats overall are in decline across the U.S., primarily from development; over a 200-year period from the 1780s to 1980s, the contiguous U.S. lost an estimated 53% of original wetlands [66]. More recently, although the pace of loss appears to have slowed [67], wetland loss continues to occur at a high rate in certain regions. For example, the northeast and southeast regions of the U.S., where firefly species richness is highest (Fig 3A), both saw downward trends in wetlands acreage from 1992 to 2010 [68]. Coastal regions are particularly at risk; an estimated 80,000 acres of coastal wetlands in the contiguous U.S. are lost each year due to development, drainage, storms, and sea level rise [67]. Loss of wetland habitat due to sea level rise was identified as a major threat to coastal firefly species like *Photuris bethaniensis* and *Micronaspis floridana*. Because these are habitat specialists and occupy small areas threatened by intense coastal development, their opportunity to disperse to other sites is limited [69].

Development is also linked to light pollution, or artificial light at night (ALAN), a threat affecting 17 out of the 18 threatened species in this study. ALAN is comprised of skyglow (the diffuse glowing haze over populated areas), glare (excessive amounts of lighting), and light trespass (light that spills out beyond its intended target). It can be caused by a number of different sources, from commercial and residential development to vehicle headlights and gas

**Table 2. Ecology and life history characteristics of 18 threatened firefly species in the U.S. and Canada.**

| Species name | Common name | Category | Criteria | Range | EOO (km2) | Courtship signal | Courtship activity period | Females | Habitat association | Habitat description |
|---|---|---|---|---|---|---|---|---|---|---|
| *Bicellonycha wickershamorum* | Southwest spring firefly | VU | B1ab(iii) | Arizona | 2,113–15,941 | Flash | Crepuscular | Winged | Possible specialist | Montane seeps and marshes along permanent streams |
| *Bicellonycha wickershamorum piceum* | Gila Southwest spring firefly | EN | B2ab(iii) | Arizona | Unknown | Flash | Crepuscular | Winged | Possible specialist | Montane seeps along permanent streams |
| *Bicellonycha wickershamorum wickershamorum* | Southwest spring firefly | VU | B1ab(iii) | Arizona | 2,113–9,636 | Flash | Crepuscular | Winged | Possible specialist | Montane seeps and marshes along permanent streams |
| *Lucidota luteicollis* | Florida scrub dark firefly | VU | B1ab(iii) | Florida | 13,035 | None | Diurnal | Flightless | Specialist | Upland ridges within scrub, sandhill, and pine savannah |
| *Micronaspis floridana* | Florida intertidal firefly | EN | B2ab(i,ii,iii) | Florida, Bahamas | 109,494 | Flash | Nocturnal | Winged | Specialist | Salt marshes, mudflats, and mangroves in coastal areas |
| *Photinus acuminatus* | Pointy-lobed firefly | EN | B2ab(i,ii,iii, iv,v) | Florida, Georgia, Mississippi, North Carolina, Ohio | Unknown | Flash | Nocturnal | Winged | Specialist | Bogs and marshes |
| *Photinus knulli†* | Southwest synchronous firefly | VU | B1ab(iii) | Arizona, Mexico | 8,329 | Flash | Nocturnal | Winged | Specialist | Marshes along permanent streams |
| *Photuris bethaniensis* | Bethany Beach firefly | CR | B1ab(i,ii,iii, v) | Delaware | 33 | Flash | Nocturnal | Winged | Specialist | Interdunal freshwater swales |
| *Photuris cinctipennis* | Belted firefly | EN | B1ab(ii,iii) +2ab(ii,iii) | Delaware, Maryland | 4,643 | Flash | Nocturnal | Winged | Possible specialist | Moist lowland areas within hardwood forests |
| *Photuris flavicollis* | Sky island firefly | VU | B1ab(iii) | New Mexico, Texas | 8,497 | Flash | Nocturnal | Winged | Possible specialist | Montane seeps and springs |
| *Photuris forresti†* | Loopy five firefly | EN | B1ab(i,ii,iii, iv)+2ab(i,ii, iii,iv) | South Carolina, Tennessee | 3,349 | Flash | Nocturnal | Winged | Specialist | Marshes |
| *Photuris mysticalampas* | Mysterious lantern firefly | EN | B1B2ab(ii, iii) | Delaware | 1,050 | Flash | Nocturnal | Winged | Specialist | Forested peatland floodplains |
| *Photuris pensylvanica* | Dot-dash firefly | VU | B2ab(iii) | Delaware, Maryland, New Jersey, New York, Pennsylvania | 14,023–86,276 | Flash | Nocturnal | Winged | Specialist | Tidal and non-tidal freshwater wetlands |
| *Photuris pyralomima* | None | EN | B1ab(i,ii,iii) +B2ab(i,ii, iii) | Delaware, New York | 2,285 | Flash | Nocturnal | Winged | Possible specialist | Moist grassland or shrubland |
| *Photuris walldoxeyi* | Cypress firefly | VU | B2ab(iii) | Illinois, Indiana, Mississippi, Tennessee | 69,962 | Flash | Nocturnal | Winged | Specialist | Cypress swamps |
| *Pleotomodes needhami†* | Ant-loving scrub firefly | EN | B1ab(iii) | Florida | 1,616 | Glow | Nocturnal | Flightless | Specialist | Upland ridges within xeric pine and oak scrub forests |
| *Pyractomena ecostata* | Keel-necked firefly | EN | B2ab(i,ii,iii) | Alabama, Delaware, Florida, New Jersey | 955,697 | Flash | Nocturnal | Winged | Specialist | Brackish tidal marshes |
| *Pyractomena vexillaria†* | Amber comet | EN | B2ab(i,iii) | Texas, Mexico | 32,716 | Flash | Nocturnal | Winged | Specialist | River basins within semi-arid shrubland |

† Indicates species that have been submitted to the IUCN Red List but have not yet been published. Categories and criteria for these species are thus considered pending until formal publication.

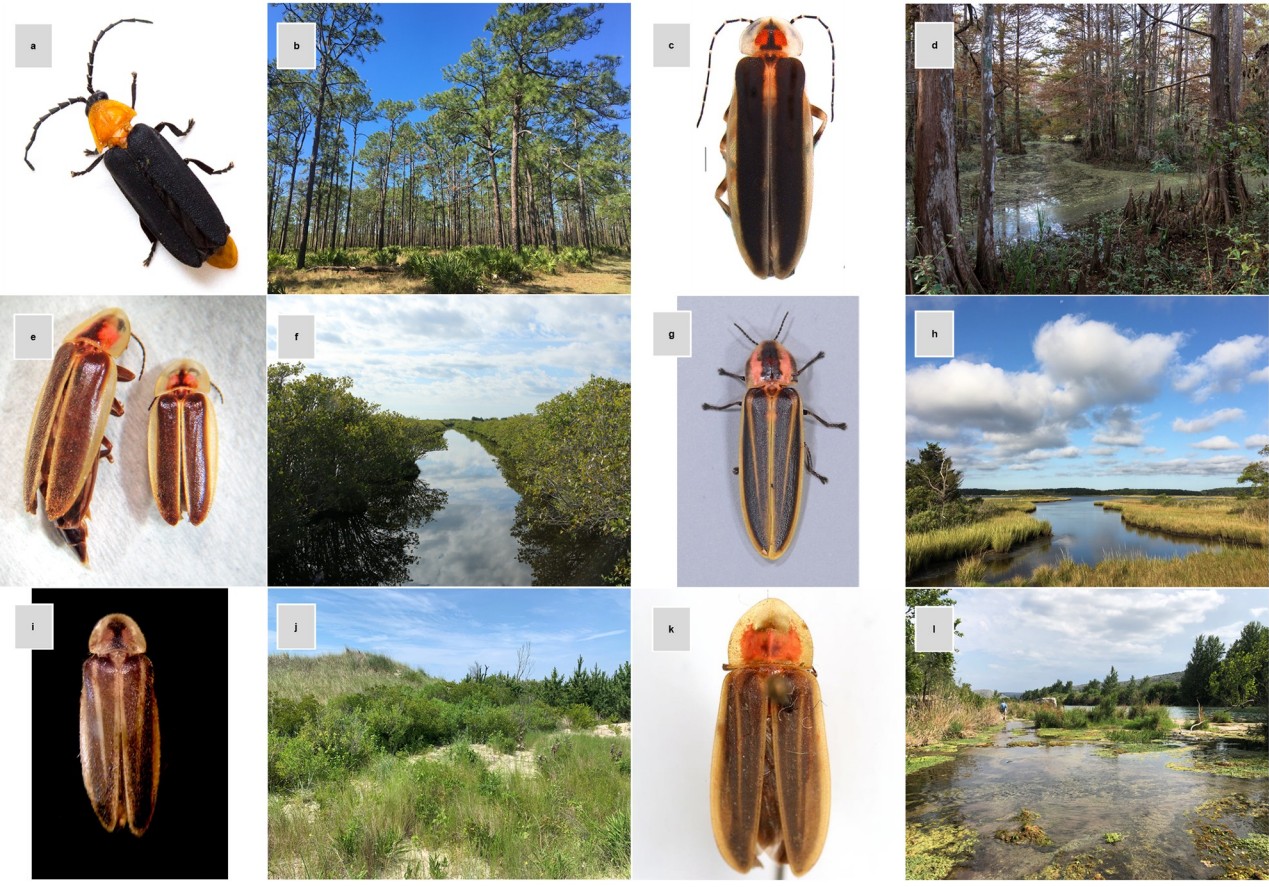

**Fig 2. Threatened fireflies tend to be restricted to specialized habitats.** (a) *Lucidota luteicollis*. Printed with permission from Brandon Woo, CC BY 4.0. (b) Characteristic upland sand scrub habitat of *L. luteicollis* in Florida. Reprinted from Leo Miranda/USFWS [56], CC BY 2.0. (c) *Photuris walldoxeyi*. Printed with permission from Luiz Silveira, CC BY 4.0. (d) Cypress swamp characteristic of *P. walldoxeyi* habitat. Reprinted from capt_tain Tom [57], CC BY 2.0. (e) *Micronaspis floridana*. Printed with permission from Lynn Faust, CC BY 4.0. (f) Coastal salt marsh typical of *M. floridana* in Cedar Key, Florida. Reprinted from Karen Kleis [58], CC BY 2.0. (g) *Pyractomena ecostata*. Printed with permission from Oliver Keller, CC BY 4.0. (h) Atlantic tidal marsh characteristic of *P. ecostata* habitat in Delaware. Reprinted from Andy Atzert [59], CC BY 2.0. (i) *Photuris bethaniensis*. Printed with permission from Christopher M. Heckscher, CC BY 4.0. (j) Interdunal swale habitat characteristic of *P. bethaniensis* in Delaware. Printed with permission from Emily May, CC BY 4.0. (k) *Pyractomena vexillaria*. Printed with permission from Mike Quinn, CC BY 4.0. (l) *P. vexillaria* habitat along the Devils River, Texas. Printed with permission from Ben Pfeiffer, CC BY 4.0.

flares. All sources of ALAN have the potential to drive firefly population declines. More than 75% of firefly species in the United States and Canada are nocturnally active or crepuscular species that utilize bioluminescent courtship signals that are sensitive to environmental light conditions. A growing body of research suggests that artificial light from street lamps, residences, and other sources may impede the ability of males to locate female mates [23,73]. For example, experimental studies have shown that artificial light can interfere with the production and reception of courtship signals [74,75] and inhibit larval dispersal [76], which could affect reproductive fitness and have cascading impacts for firefly populations.

## Species distributions

Fireflies were recorded in every U.S. state except for Hawaii and every Canadian province and territory except Nunavut (Fig 3A; S1 Table). Thirty species (23%) were thought to be endemic to a single state or province (27% of such species were categorized as threatened). States that

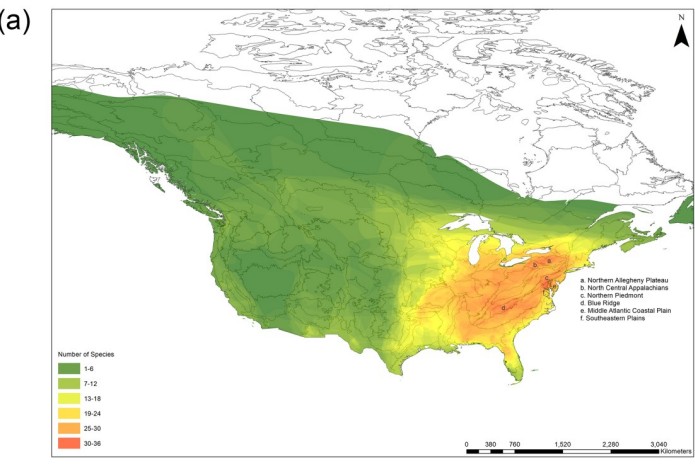

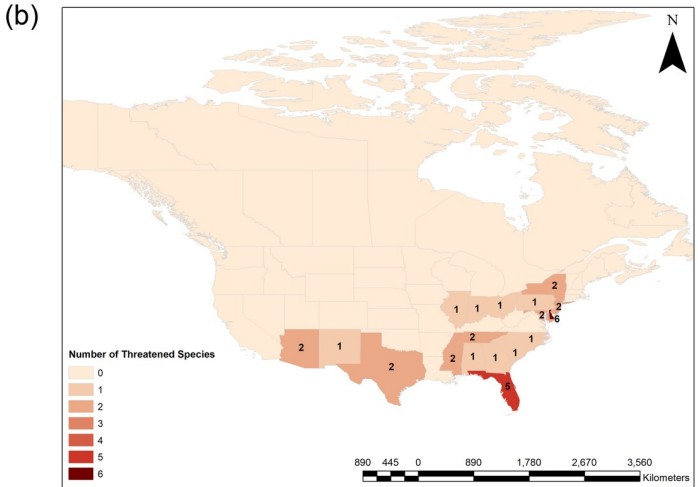

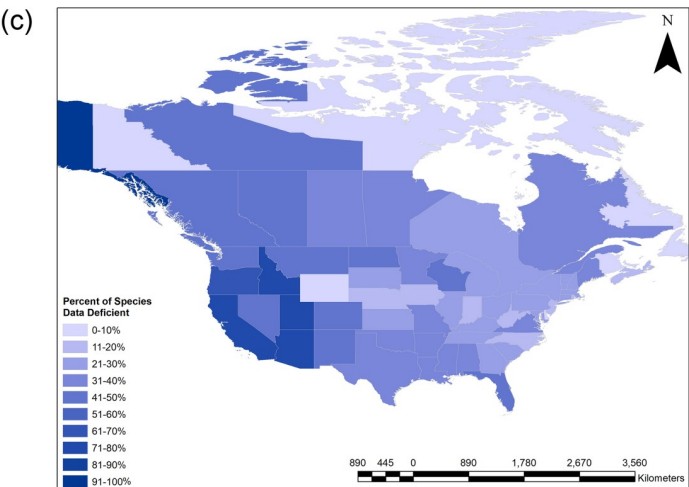

**Fig 3. Species distributions and status summaries.** (A) Overall species richness of fireflies in the U.S. and Canada. Gray lines indicate Level III Ecoregion boundaries. Ecoregions with the highest species richness are labeled; all others are unlabeled. (B) Geographic summary of threatened (CR, EN, or VU) firefly species. Note that one of the taxa indicated in Arizona consists of 2 subspecies that are also threatened. (C) Geographic summary of Data Deficient firefly species, shown as a percent of the total number of species reported from each state. The maps were created by

the authors based on our Red List assessment results using ArcMap by Esri [70]. Public domain administrative boundary layers were obtained from Natural Earth [71]. The Fig 3A Level III Ecoregions layer was obtained from the U.S. Environmental Protection Agency (EPA) [72].

support the highest numbers of endemic species include Arizona (eight species), Florida (eight species), California (five species), and Texas (four species) (S1 Table). In general, species richness increases moving from west to east; when we overlaid Level III Ecoregion [72] boundaries on the map, the major hotspots of species richness (defined here as areas with more than 30 species across most of the ecoregion) were in the North Central Appalachians, Northern Allegheny Plateau, Northern Piedmont, and Blue Ridge ecoregions (Fig 3A). The Middle Atlantic Coastal Plain and Southeastern Plains also support more than 30 species each, but only across a small part of the ecoregion. Threatened species are concentrated in the Mid-Atlantic and Southeast regions (Fig 3B), while DD species are scattered throughout the two countries (Fig 3C). All 18 threatened species have narrow geographic ranges, with 10 thought to be endemic to a single state. It is likely that these distributions are heavily influenced by sampling bias and geographic concentrations of species experts; for example, West Virginia likely has higher species richness than is currently reported (12 species) given the high number of species found in surrounding states, but sampling efforts are not yet as comprehensive in this state.

## Moving forward: Conservation actions

The results of our assessments have made it clear that additional conservation actions are needed for fireflies in the U.S. and Canada (S3 Table). More specifically, this includes identifying and protecting populations of at-risk species, preserving and restoring firefly habitat, gathering data on population trends, and filling critical information gaps for those Data Deficient species suspected to be at risk. Science communication is also important: conducting education and outreach can help ensure that fireflies and their needs are taken into consideration. In the following sections, we expand on these recommended next steps to prevent firefly species extinctions.

### Protect at-risk species

These assessments identified 18 taxa at risk of extinction and two others that may be at risk in the near future (Near Threatened). Currently, very few conservation measures are in place to protect North American fireflies. Species-specific conservation actions should focus on prioritizing these threatened species. The Critically Endangered Bethany Beach firefly, *Photuris bethaniensis*, which is listed as State Endangered in Delaware, is currently under consideration for Endangered Species Act (ESA) listing—the first firefly to be petitioned [77]. No other fireflies are included in endangered species lists for any state or province, and no regulatory mechanisms are in place to protect at-risk species. However, several states, including Delaware, Florida, Indiana, Maryland, and South Carolina, do include at-risk firefly species as Species of Greatest Conservation Need (SGCN) in their State Wildlife Action Plans. These plans are intended to inform conservation priorities and actions at the state level, with a particular focus on strategies for managing and protecting SGCN. We recommend that state wildlife agencies include threatened firefly species as SGCN, if they are not already included. Furthermore, we suggest that Data Deficient species that we suspect to be threatened also be considered as SGCN (S4 Table).

### Preserve and restore habitat

Because habitat loss is a key threat to fireflies, preserving and restoring habitat for threatened firefly species will be an integral part of any conservation efforts. This can be accomplished in several ways:

- Protect and restore occupied and adjacent habitat to support threatened species, such as coastal salt marshes and cypress swamps (see Table 2)

- Work with local conservation organizations and land trusts to establish Firefly Sanctuaries in areas with threatened or endemic species or high species diversity

- Work with tourism sites to establish and implement clear guidelines for managers, tour operators, and visitors to ensure that fireflies are protected from tourism-related threats [e.g., 17]

- Mitigate light pollution close to firefly habitat through educational outreach programs, Dark Sky Initiatives [78], and updates to city lighting ordinances

- Refrain from using pesticides, particularly insecticides and molluscicides, in areas used by fireflies, as these can kill fireflies and their prey, and may have other unintended consequences

### Survey and monitor populations

Surveys and monitoring were identified as key conservation actions for all 18 threatened species and all of the Data Deficient species. A shortage of survey efforts and population monitoring for the majority of species—due in no small part to a lack of standardized methodology for tracking them, short species activity windows, difficulty in reaching survey sites, difficulty in identification at the species level, the hazards posed by nocturnal fieldwork, and a general lack of funding—severely limits our ability to track firefly populations over time. Baseline inventories to determine species distributions are needed to better understand the conservation status and needs of individual species. In particular, randomized grid surveys over large geographic areas, coupled with targeted surveys for known threatened species, could help reduce survey bias and increase the scope of survey efforts. Even with temporal and spatial biases, however, it is possible to extract meaningful results that can inform conservation efforts [e.g., 79–81].

While such surveys at the local, state, and federal levels are recommended, successful survey programs may rest on integrating these efforts with large-scale initiatives across wide geographic areas. Community ("citizen") science projects have a long and illustrious history of engaging public interest while benefiting conservation efforts, and many have effectively incorporated web-based tools [e.g., 82] to increase participation and dissemination of data. For many insects, incorporating species-level identifications in such projects can be challenging due to insects' hyper diversity, small size, often abstruse taxonomy, and difficulty of field identification. In the face of recent insect declines, however, community science has the potential to fulfill a critical need in documenting species distributions and population trends [26,83,84]. For fireflies in particular, Firefly Watch [85] has engaged thousands of community scientists across North America since it started in 2008, and additional resources are now available to aid field-based species identifications [28]. Regional projects such as the Western Firefly Project [86] are also filling data gaps. To gather additional occurrence data for species not typically covered by these programs, volunteers who are trained to identify their local threatened and

Data Deficient species could contribute high-quality, geotagged photographs and details of flash pattern behavior, when applicable, to iNaturalist [87].

Given the apparent rarity or limited abundance of some firefly populations, large-scale collecting should be avoided. Lethal sampling is generally not recommended other than for the purposes of collecting voucher specimens for museums to verify species occurrence. When possible, geotagged voucher photos with corresponding habitat and behavior information should be used in lieu of physical vouchers.

## Fill data gaps

More than half (53%) of the North American firefly species assessed were Data Deficient, indicating that more information is needed to estimate these species' extinction risks. Data Deficient species tended to be characterized by cryptic life histories, non-flashing communication behavior, or flightless adult females. For example, a large portion of glow-worm species (79%) and diurnal fireflies (68%) are categorized as Data Deficient, as opposed to 38% of flashing species (S5 Table). The comparatively high rate of data deficiency in glow-worm species is likely due in part to the difficulty in detecting these less conspicuous species. Glow-worm species have flightless females, adult males in most of these species do not produce light. Combined with their nocturnal activity period, diminutive body size, and inconspicuous female light signals, many of these are easily overlooked. This underscores the need for specialized survey protocols and additional research into firefly species that do not use flash signals in courtship, particularly basic life history studies that examine habitat associations and microhabitat needs, larval and adult diets, activity periods, and threats. Details about priority Data Deficient species can be found in S1 File.

## Engage and educate

Effective science communication can play an important role in achieving conservation goals by garnering public support, attracting funding, driving policy changes, and promoting informed decision making [88–93]. For small yet charismatic animals like fireflies, building up communication efforts may lead to increased support for not only fireflies and their habitats, but invertebrate conservation more broadly. In tandem with the conservation actions discussed here, we recommend increasing outreach and education efforts to share new findings and facilitate collaboration. In addition to community science initiatives, which can be an effective means for increasing engagement with conservation [94,95], creative use of workshops, social and popular media, school fieldtrips, and museum exhibits, among others, may help to build support for firefly conservation. Development and distribution of science-based conservation guidelines [e.g., 96] and other educational resources can also be helpful in outreach efforts.

## Conclusions

This paper summarizes the first global IUCN Red List assessments for fireflies. While it does not include all described species in the U.S. and Canada, it represents a substantial step forward in understanding extinction risk for North American species. We now have a foundation from which we can work, which spans the setting of conservation priorities to the establishment of a baseline against which future findings can be compared. We hope the results and implications discussed in this paper will catalyze action to study and conserve fireflies, not just in the U.S. and Canada but everywhere fireflies are found.

## Supporting information

**S1 Table. Firefly species distributions in the U.S. and Canada.** Species name: †: Awaiting publication on the IUCN Red List. RL category: DD*: Potentially threatened DD species. Occurrence and Distribution: Extant (E): Species has been reported since 2000; Presence uncertain (U): Species reported prior to 2000; Possibly Extant (PE): No known records but habitat or locality is appropriate and species may occur here; Possibly Extinct (PX): Species has not been seen in many years despite comprehensive survey efforts.
(XLSX)

**S2 Table. Firefly taxonomic experts consulted in this project.** Contributor: Contributed information to Red List assessments; Assessor: Co-authored Red List assessments; Reviewer: Reviewed Red List assessments; Determiner: Verified species IDs.
(XLSX)

**S3 Table. Recommended conservation actions for threatened firefly species in the U.S. and Canada.**
(XLSX)

**S4 Table. Potentially threatened DD firefly species.**
(XLSX)

**S5 Table. North American firefly genera with total number of species assessed, percentage of threatened and data deficient species, and behavioral type.**
(XLSX)

**S1 File.**
(DOCX)

## Acknowledgments

We thank Ron Lyons, Laura Hughes, and Jason Davis for contributing to the Red List assessments; Cheryl Mollohan for sharing data on Arizona fireflies; Annie Nguyen and Aubrey Hornor for assisting with data compilation and georeferencing; Clay Meredith for co-facilitating the IUCN Red List training workshop; the dedicated staff at the Red List Unit, in particular Janet Scott, for publishing the Red List assessments; and two anonymous reviewers for improving this manuscript. We are grateful to the many firefly researchers past and present who have enhanced our understanding of North American fireflies, the vast network of community scientists who share their observations with the scientific community, and the many photographers who contributed their photos to the Red List publications and this paper.

## Author Contributions

**Conceptualization:** Candace E. Fallon, Anna C. Walker, Sara Lewis, Sarina Jepsen.

**Investigation:** Candace E. Fallon, Anna C. Walker, Sara Lewis, Joseph Cicero, Lynn Faust, Christopher M. Heckscher, Cisteil X. Pérez-Hernández, Ben Pfeiffer.

**Writing – original draft:** Candace E. Fallon, Anna C. Walker, Sara Lewis.

**Writing – review & editing:** Candace E. Fallon, Anna C. Walker, Sara Lewis, Joseph Cicero, Lynn Faust, Christopher M. Heckscher, Cisteil X. Pérez-Hernández, Ben Pfeiffer, Sarina Jepsen.

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
