## [Decision Letter · Decision Letter 0]

4 Aug 2021

PONE-D-21-17592

Evaluating firefly extinction risk: Initial Red List assessments for North America

PLOS ONE

Dear Dr. Fallon,

Thank you for submitting your manuscript to PLOS ONE. After careful consideration, we feel that it has merit but does not fully meet PLOS ONE’s publication criteria as it currently stands. Therefore, we invite you to submit a revised version of the manuscript that addresses the points raised during the review process.

We look forward to receiving your revised manuscript.

Kind regards,

Daniel de Paiva Silva, Ph.D.

Academic Editor

PLOS ONE

Journal Requirements:

3. We noted in your submission details that a portion of your manuscript may have been presented or published elsewhere. [Individual species information is published on the IUCN Red List of Threatened Species (https://www.iucnredlist.org/).] Please clarify whether this publication was peer-reviewed and formally published. If this work was previously peer-reviewed and published, in the cover letter please provide the reason that this work does not constitute dual publication and should be included in the current manuscript.

4. We note that Figures #1A, 1B and 1C in your submission contain [map/satellite] images which may be copyrighted. All PLOS content is published under the Creative Commons Attribution License (CC BY 4.0), which means that the manuscript, images, and Supporting Information files will be freely available online, and any third party is permitted to access, download, copy, distribute, and use these materials in any way, even commercially, with proper attribution. For these reasons, we cannot publish previously copyrighted maps or satellite images created using proprietary data, such as Google software (Google Maps, Street View, and Earth). For more information, see our copyright guidelines: http://journals.plos.org/plosone/s/licenses-and-copyright.

a. You may seek permission from the original copyright holder of Figures #1A, 1B and 1C to publish the content specifically under the CC BY 4.0 license.  

5. We note that Figure #2 in your submission contain copyrighted images. All PLOS content is published under the Creative Commons Attribution License (CC BY 4.0), which means that the manuscript, images, and Supporting Information files will be freely available online, and any third party is permitted to access, download, copy, distribute, and use these materials in any way, even commercially, with proper attribution. For more information, see our copyright guidelines: http://journals.plos.org/plosone/s/licenses-and-copyright.

a. You may seek permission from the original copyright holder of Figure #2 to publish the content specifically under the CC BY 4.0 license. 

Additional Editor Comments:

Dear Fallon et al.,

After the first review round, I am pleased to inform to you that your study received two "minor reviews" status by two different reviewers. Therefore, your manuscript is practically there concerning its acceptance in PLoS One. As you will see that the amount of improvements to be done are few, please consider resubmitting your study by September 5th, 2021. Do not hesitate to resubmit earlier if you are able to. In case you need more time, please let me know. Finally, by the time you resubmit, do not forget to prepare a rebuttal letter, where you will explain all the performed changes and justify to the reviewers the reason why any given change was not done.

Once again, congratulations!

Sincerely,

Daniel Silva, Ph.D.

Reviewers' comments:

Reviewer's Responses to Questions

**Comments to the Author**

1. Is the manuscript technically sound, and do the data support the conclusions?

Reviewer #1: Yes

Reviewer #2: Yes

2. Has the statistical analysis been performed appropriately and rigorously? 

Reviewer #1: Yes

Reviewer #2: N/A

3. Have the authors made all data underlying the findings in their manuscript fully available?

Reviewer #1: Yes

Reviewer #2: Yes

4. Is the manuscript presented in an intelligible fashion and written in standard English?

Reviewer #1: Yes

Reviewer #2: Yes

5. Review Comments to the Author

Reviewer #1: Comments and reviewer recommendation for manuscript number PONE-D-21-17592 “Evaluating firefly extinction risk: Initial Red List assessments for North America” by Fallon et al.

This manuscript deals with an interesting topic, namely the assessement of fireflies against the IUCN criteria in order to compile the first Red List of this charismatic beetle group globally. Despite a number of drawbacks (relatively few data on trends and distribution), the authors managed to compile an initial list of most threatened fireflies in North America. This resulted in the use of only one out of five possible IUCN criteria (i.e. geographical range) to classify species in the different Red List categories. Since this criterion does not only use rarity as such, but demands at least two out of three subcriteria to be fulfilled (e.g. fragmentation, decline), it is not clear how the authors evaluated these additional criteria. This should be explained in a more detailed way so that future Red Lists of this or other less well-known insect groups can learn from and repeat the method used in this exercise. Apart from that, the manuscript is well written and reads easily, with clear figures and ample and appropriate references. I would replace Table 1 with a graph and mention the numbers in the text itself, because a graph is much more informative than numbers in a table. Some additional comments are given in the attached pdf. Nice manuscript!

Reviewer #2: Insect declines are a high conservation priority, but poorly understood, particularly outside of some butterfly species. Consequently, it is very exciting to see this extensive foray into firefly extinction risk using the IUCN criteria.

Great justification for the work, excellent data, strong support from taxonomic experts, and effective application of IUCN criteria. Well-presented results and focused discussion. The paper ends with an excellent road map of proposed actions for conservation.

Results

- Table 2 – Since most people are not conversant with the nuances of the IUCN sub-criteria, and it would be time-consuming for folks to walk through IUCN documents to figure it out, it would be helpful for readers if a column was added that used text to explain the criteria; or, since there is a lot of overlap across species and few sub-criteria used in the listing, add multiple columns listing the sub-criteria, with checkmarks for each species where they apply.

- Also, the text includes common names along with the scientific names – consider adding them to Table 2.

Minor comments

- Line 26 ‘threat of extinction from threats’ is awkward

- Lines 25-26 – by ‘under threat’, do you mean in categories CR, EN, and VU? Better to be specific with categories – this is clarified on line 140

- Line 27 – are the unevaluated additional 21% of the firefly species in the U.S. & Canada also DD, and that was why they weren’t evaluated? Seems to be suggested on line 117. If yes, do you want to increase this value, or are there IUCN rules restricting use of the term?

- Lines 120-125 – any information on databases searched and search terms used would help readers evaluate/understand the thoroughness of the search

- Line 245 – instead of ‘capacity’, it might be more accurate to say ‘opportunity’ (they might be capable of dispersing, but there is nowhere to go)

- Line 247 ’17 of 18 of the at-risk species in our study’ – not a generic frequency across all firefly species

- Line 269 ‘currently, currently’

- Line 281 – I’m not sure what ‘converting the assessments’ means

- Line 296 – please provide a citation for ‘Dark Sky Initiatives’

- Line 327 – it might be worth adding ‘for museums’ to ‘voucher specimens’

- Section starting line 343 – I’m not sure all engagement and education leads to action – it would be good to include a few citations in this section of examples where it has been shown to be effective.

6. PLOS authors have the option to publish the peer review history of their article (what does this mean?). If published, this will include your full peer review and any attached files.

Reviewer #1: No

Reviewer #2: No

---

## [Author Response · Author response to Decision Letter 0]

3 Sep 2021

[Please note that our response has also been uploaded (with better formatting) with the other revision files.]

Response to Reviewers

Below, you will find our responses to the editorial and reviewer comments, including point-by-point descriptions of changes made.

Reviewer #1: Comments and reviewer recommendation for manuscript number PONE-D-21-17592 “Evaluating firefly extinction risk: Initial Red List assessments for North America” by Fallon et al.

This manuscript deals with an interesting topic, namely the assessment of fireflies against the IUCN criteria in order to compile the first Red List of this charismatic beetle group globally. Despite a number of drawbacks (relatively few data on trends and distribution), the authors managed to compile an initial list of most threatened fireflies in North America. This resulted in the use of only one out of five possible IUCN criteria (i.e. geographical range) to classify species in the different Red List categories. Since this criterion does not only use rarity as such, but demands at least two out of three subcriteria to be fulfilled (e.g. fragmentation, decline), it is not clear how the authors evaluated these additional criteria. This should be explained in a more detailed way so that future Red Lists of this or other less well-known insect groups can learn from and repeat the method used in this exercise. Apart from that, the manuscript is well written and reads easily, with clear figures and ample and appropriate references. I would replace Table 1 with a graph and mention the numbers in the text itself, because a graph is much more informative than numbers in a table. Some additional comments are given in the attached pdf. Nice manuscript!

Response: Thank you! Regarding the comment on evaluation of subcriteria, we have:

1. Added a reference to the IUCN Red List Guidelines (line 158), which provide more background on the application of subcriteria during the Red List process, and 

2. Added details to the S1 Supporting Information regarding how the Criterion B subcriteria were evaluated and applied in this particular study. These changes can be viewed in the ‘Revised S1 Supporting Information with Tracked Changes.’

In addition, per the reviewer’s suggestion, we shortened Table 1 to show only the summary information, created a new graph (Fig 1) depicting percentage of species in each Red List category, and added the numbers in the text itself.

Comments provided in PONE-D-21-17592_reviewer1:

Line 118: Replaced ‘species’ with ‘taxa’

Line 148: Changed ‘survive only captivity’ to ‘survive only in captivity’

Comment line 151: Discuss maybe how the availability of additional criteria (beyond criterion B) might have affected the Red list assessment. More or less species in threatened categories?

Response: We debated adding this but were not sure that it added much to the discussion. The availability of additional criteria would not reduce the number of species in threatened categories, but if we had additional data on population size and trend, DD species could be assessed against the criteria and some would likely come out as threatened. In addition, some threatened species might fall into higher threat categories with more data. This is probably true of many invertebrate species, which lack population data.

Comment line 181: This paragraph should come first in the results section! In what is now the first paragraph, you mention that 18 species are threatened, but the reader does not know that yet ... This is also the main subject of this manuscript so it is good to start the Results section.

Response: We agree. We have rearranged the order of the Results and Discussion so that Extinction Risk and Threats comes before the Species Distributions subsection.

Lines 183-185: Added number of species for each Red List category

Line 189: Capitalized ‘Data Deficient’

Comment line 192 regarding Table 1: I would suggest to turn this table into a graph and to add the numbers to the text itself. Visualising proportions is much clearer than numbers in a table, in my opinion.

Response: We replaced the first part of Table 1 with a graph (Figure 1) but kept the second part of the table that provides the summary information.

Line 269: Deleted repeated word ‘currently’

Line 280: Changed ‘DD’ to “Data Deficient’

Comment line 282: This last sentence is only known to American readers but foreign scientists do not have any idea what the NatureServe ranking en de NatureServe Network is. The authors should either omit this or give some explanation about it ...

Response: We deleted this, in part because these assessments are now underway, and so this recommendation is in the process of being met.

Line 302: Changed ‘DD’ to “Data Deficient’

Comment line 309: At present, quite some statistical techniques allow to correct for both temporal and spatial biases in survey effort (e.g. Isaac et al. (2014) Statistics for citizen science: extracting signals of change from noisy ecological data. Methods in Ecology and Evolution 5 (10): 1052-1060. https://doi.org/10.1111/2041-210X.12254).

Response: We added a sentence acknowledging these statistical techniques, citing this paper and two others in support.

Lines 322, 331, and 342: Capitalized ‘Data Deficient’

Reviewer #2: 

Insect declines are a high conservation priority, but poorly understood, particularly outside of some butterfly species. Consequently, it is very exciting to see this extensive foray into firefly extinction risk using the IUCN criteria. Great justification for the work, excellent data, strong support from taxonomic experts, and effective application of IUCN criteria. Well-presented results and focused discussion. The paper ends with an excellent road map of proposed actions for conservation.

Response: Thank you!

Results

- Table 2 – Since most people are not conversant with the nuances of the IUCN sub-criteria, and it would be time-consuming for folks to walk through IUCN documents to figure it out, it would be helpful for readers if a column was added that used text to explain the criteria; or, since there is a lot of overlap across species and few sub-criteria used in the listing, add multiple columns listing the sub-criteria, with checkmarks for each species where they apply.

Response: We agree that more information explaining the subcriteria would be helpful. Based on this comment and another comment from Reviewer #1, we have added definitions and more details about the application of subcriteria to the S1 Supporting Information. We decided this would keep the table clean while allowing those with interest to take a deeper dive into the supplemental materials. 

- Also, the text includes common names along with the scientific names – consider adding them to Table 2.

Response: Common names are provided in the second column of Table 2.

Minor comments

- Line 26 ‘threat of extinction from threats’ is awkward

Response: Changed to “…threatened with extinction (e.g. categorized as Critically Endangered, Endangered, or Vulnerable) due to various pressures…”

- Lines 25-26 – by ‘under threat’, do you mean in categories CR, EN, and VU? Better to be specific with categories – this is clarified on line 140

Response: Yes, that is what we meant. We have clarified this by adding the specific categories in parentheses.

- Line 27 – are the unevaluated additional 21% of the firefly species in the U.S. & Canada also DD, and that was why they weren’t evaluated? Seems to be suggested on line 117. If yes, do you want to increase this value, or are there IUCN rules restricting use of the term?

Response: No, the remaining 21% of species in the U.S. and Canada were not assessed, and so they have not been assigned an IUCN Red List category. We have clarified this in the manuscript (lines 118-120). We have also corrected the language in lines 28-30 to reflect the fact that the DD species were in fact assessed, but could not be fully evaluated given the lack of data (hence the DD categorization).

- Lines 120-125 – any information on databases searched and search terms used would help readers evaluate/understand the thoroughness of the search

Response: Added details about source databases and search terms.

- Line 245 – instead of ‘capacity’, it might be more accurate to say ‘opportunity’ (they might be capable of dispersing, but there is nowhere to go)

Response: Good point. Replaced ‘capacity’ with ‘opportunity.’

- Line 247 ’17 of 18 of the at-risk species in our study’ – not a generic frequency across all firefly species

Response: Clarified this statement to read “Development is also linked to light pollution, or artificial light at night (ALAN), a threat affecting 17 out of the 18 threatened species in this study.”

- Line 269 ‘currently, currently’

Response: Removed second use of ‘currently’

- Line 281 – I’m not sure what ‘converting the assessments’ means

Response: We removed the two sentences referencing NatureServe per Reviewer #1’s comment and because this process of assessing species according to NatureServe criteria is now underway.

- Line 296 – please provide a citation for ‘Dark Sky Initiatives’

Response: A citation for the Dark-Sky Association has been added as an example

- Line 327 – it might be worth adding ‘for museums’ to ‘voucher specimens’

Response: Added ‘for museums’

- Section starting line 343 – I’m not sure all engagement and education leads to action – it would be good to include a few citations in this section of examples where it has been shown to be effective

Response: Added several citations to support the statements in this section

Response: We have gone through the style requirements and applied any necessary changes, including updating the file names. All figures were processed through PACE to ensure formatting fits PLOS ONE requirements.

Response: There are no grant numbers associated with our funding sources. We have provided websites for funders that have them and updated the grant information to the following:

CEF and SJ were funded by the Samuel Freeman Charitable Trust, the Edward Gorey Charitable Trust https://edwardgorey.org/, the New-Land Foundation (http://newlandfoundation.org/), Morningstar Foundation (http://morningstarfoundation.com/), and Xerces Society members. ACW was funded by the New Mexico BioPark Society (https://bioparksociety.org/main/). The funders had no role in study design, data collection and analysis, decision to publish, or preparation of the manuscript.

3. We noted in your submission details that a portion of your manuscript may have been presented or published elsewhere. [Individual species information is published on the IUCN Red List of Threatened Species (https://www.iucnredlist.org/).] Please clarify whether this publication was peer-reviewed and formally published. If this work was previously peer-reviewed and published, in the cover letter please provide the reason that this work does not constitute dual publication and should be included in the current manuscript.

Response: This work does not constitute dual publication and it should be included in the current manuscript. As stated in the above cover letter, this study, if published in PLOS ONE, would be the first peer-reviewed journal article summarizing and interpreting the results of 132 North American firefly species assessed for the IUCN Red List of Threatened Species. While the assessments themselves are peer-reviewed, each assessment represents a standalone single-species summary. Therefore, this paper constitutes a new publication broadly focused on the status of the North American firefly fauna.

4. We note that Figures #1A, 1B and 1C in your submission contain [map/satellite] images which may be copyrighted. All PLOS content is published under the Creative Commons Attribution License (CC BY 4.0), which means that the manuscript, images, and Supporting Information files will be freely available online, and any third party is permitted to access, download, copy, distribute, and use these materials in any way, even commercially, with proper attribution. For these reasons, we cannot publish previously copyrighted maps or satellite images created using proprietary data, such as Google software (Google Maps, Street View, and Earth). For more information, see our copyright guidelines: http://journals.plos.org/plosone/s/licenses-and-copyright.

a. You may seek permission from the original copyright holder of Figures #1A, 1B and 1C to publish the content specifically under the CC BY 4.0 license. 

Response: We revised Figures #1A, 1B and 1C (now labeled 3A, 3B, and 3C) and have provided replacement figures that comply with the CC BY 4.0 license. We have also updated the figure captions with new source information. 

5. We note that Figure #2 in your submission contain copyrighted images. All PLOS content is published under the Creative Commons Attribution License (CC BY 4.0), which means that the manuscript, images, and Supporting Information files will be freely available online, and any third party is permitted to access, download, copy, distribute, and use these materials in any way, even commercially, with proper attribution. For more information, see our copyright guidelines: http://journals.plos.org/plosone/s/licenses-and-copyright.

a. You may seek permission from the original copyright holder of Figure #2 to publish the content specifically under the CC BY 4.0 license. 

Response: We revised Figure #2 and have provided a replacement figure with an updated figure caption that complies with the CC BY 4.0 license. In addition, we have secured Content Permission Forms for all copyrighted images and uploaded them as “Other” files with our submission. For photos that are publicly available with CC BY licenses (i.e., the photos we sourced from Flickr), we did not obtain Content Permission Forms but do note in the caption that the images are licensed under CC BY.

Response: We reviewed the reference list for completion and accuracy. We also added the following citations to support revisions elsewhere in the paper:

[46] Scientific Literature. In: Fireflyers International Network [Internet]. [cited 9 Aug 2021]. Available: https://fireflyersinternational.net/scientific-literature.

[56] Leo Miranda USF and WSS. Longleaf pine habitat. 2017. Available: https://www.flickr.com/photos/usfwssoutheast/33748135511/

[57] capt_tain Tom. DSC_1795. 2017. Available: https://www.flickr.com/photos/87744089@N08/38521089802/

[58] Kleis K. Merritt Island National Wildlife Refuge. 2014. Available: https://www.flickr.com/photos/hollykl/12811838533/

[59] Atzert A. I heard the buzz of insects, the wind in my ears, and a far off car approaching. 2017. Available: https://www.flickr.com/photos/andyatzert/36761693093/

[75] Esri Inc. ArcGIS Desktop 10.8.1. Redlands, CA: Environmental Systems Research Institute; 2020.

[76] Natural Earth - Free vector and raster map data at 1:10m, 1:50m, and 1:110m scales. [cited 18 Aug 2021]. Available: https://www.naturalearthdata.com/

[78] International Dark Sky Association. In: International Dark-Sky Association [Internet]. [cited 9 Aug 2021]. Available: https://www.darksky.org/

[79] Bird TJ, Bates AE, Lefcheck JS, Hill NA, Thomson RJ, Edgar GJ, et al. Statistical solutions for error and bias in global citizen science datasets. Biological Conservation. 2014;173: 144–154. doi:10.1016/j.biocon.2013.07.037

[80] Isaac NJB, Strien AJ van, August TA, Zeeuw MP de, Roy DB. Statistics for citizen science: extracting signals of change from noisy ecological data. Methods in Ecology and Evolution. 2014;5: 1052–1060. doi:10.1111/2041-210X.12254

[81] August T, Fox R, Roy DB, Pocock MJO. Data-derived metrics describing the behaviour of field-based citizen scientists provide insights for project design and modelling bias. Sci Rep. 2020;10: 11009. doi:10.1038/s41598-020-67658-3

[87] iNaturalist. In: iNaturalist [Internet]. [cited 25 May 2021]. Available: https://www.inaturalist.org/

[88] Bickford D, Posa MRC, Qie L, Campos-Arceiz A, Kudavidanage EP. Science communication for biodiversity conservation. Biological Conservation. 2012;151: 74–76. doi:10.1016/j.biocon.2011.12.016

[89] Scheufele DA. Science communication as political communication. PNAS. 2014;111: 13585–13592. doi:10.1073/pnas.1317516111

[90] Hall DM, Steiner R. Insect pollinator conservation policy innovations at subnational levels: Lessons for lawmakers. Environmental Science & Policy. 2019;93: 118–128. doi:10.1016/j.envsci.2018.12.026

[91] Samways MJ, Barton PS, Birkhofer K, Chichorro F, Deacon C, Fartmann T, et al. Solutions for humanity on how to conserve insects. Biological Conservation. 2020;242: 108427. doi:10.1016/j.biocon.2020.108427

[92] Saunders ME, Janes JK, O’Hanlon JC. Moving on from the insect apocalypse narrative: Engaging with evidence-based insect conservation. BioScience. 2020;70: 80–89. doi:10.1093/biosci/biz143

[93] Monarch Butterfly and Pollinators Conservation Fund. In: NFWF [Internet]. 2021 [cited 20 Aug 2021]. Available: https://www.nfwf.org/programs/monarch-butterfly-and-pollinators-conservation-fund

[94] Dickinson JL, Shirk J, Bonter D, Bonney R, Crain RL, Martin J, et al. The current state of citizen science as a tool for ecological research and public engagement. Frontiers in Ecology and the Environment. 2012;10: 291–297. doi:10.1890/110236

[95] Lewandowski EJ, Oberhauser KS. Butterfly citizen scientists in the United States increase their engagement in conservation. Biological Conservation. 2017;208: 106–112. doi:10.1016/j.biocon.2015.07.029

[96] Fallon C, Hoyle S, Lewis S, Owens A, Lee-Mäder E, Black SH, et al. Conserving the jewels of the night: Guidelines for protecting fireflies in the United States and Canada. Portland, OR: The Xerces Society for Invertebrate Conservation; 2019 p. 56.

---

## [Decision Letter · Decision Letter 1]

19 Oct 2021

Evaluating firefly extinction risk: Initial Red List assessments for North America

PONE-D-21-17592R1

Dear Dr. Fallon,

We’re pleased to inform you that your manuscript has been judged scientifically suitable for publication and will be formally accepted for publication once it meets all outstanding technical requirements.

Kind regards,

Daniel de Paiva Silva, Ph.D.

Academic Editor

PLOS ONE

Additional Editor Comments (optional):

Reviewers' comments:

Reviewer's Responses to Questions

**Comments to the Author**

1. If the authors have adequately addressed your comments raised in a previous round of review and you feel that this manuscript is now acceptable for publication, you may indicate that here to bypass the “Comments to the Author” section, enter your conflict of interest statement in the “Confidential to Editor” section, and submit your "Accept" recommendation.

Reviewer #1: All comments have been addressed

Reviewer #2: All comments have been addressed

2. Is the manuscript technically sound, and do the data support the conclusions?

Reviewer #1: Yes

Reviewer #2: Yes

3. Has the statistical analysis been performed appropriately and rigorously? 

Reviewer #1: Yes

Reviewer #2: N/A

4. Have the authors made all data underlying the findings in their manuscript fully available?

Reviewer #1: (No Response)

Reviewer #2: Yes

5. Is the manuscript presented in an intelligible fashion and written in standard English?

Reviewer #1: Yes

Reviewer #2: Yes

6. Review Comments to the Author

Reviewer #1: Comments and reviewer recommendation for manuscript number PONE-D-21-17592R1 “Evaluating firefly extinction risk: Initial Red List assessments for North America” by Fallon et al.

First of all, I would like to apologise for the delay in reviewing this manuscript.

The revised version of this manuscript has carefully incorporated, most, if not all comments of the reviewers. I have no further comments and congratulate the authors with this inspiring manuscript that will, hopefully, incite more people to assess the threat status of lesser-known invertebrate taxa.

In my opinion, this manuscript is now publishable in PLoS ONE.

Reviewer #2: (No Response)

7. PLOS authors have the option to publish the peer review history of their article (what does this mean?). If published, this will include your full peer review and any attached files.

Reviewer #1: No

Reviewer #2: No

---

## [Editor Report · Acceptance letter]

21 Oct 2021

PONE-D-21-17592R1 

Evaluating firefly extinction risk: Initial Red List assessments for North America 

Dear Dr. Fallon:

I'm pleased to inform you that your manuscript has been deemed suitable for publication in PLOS ONE. Congratulations! Your manuscript is now with our production department. 

Kind regards, 

on behalf of

Dr. Daniel de Paiva Silva 

Academic Editor

PLOS ONE